# Unilateral Exudative Retinal Detachment and Uveitis Accompanied by a Large Subretinal Deposit on the Macula Secondary to Bartonella Henselae Infection

**DOI:** 10.3390/diagnostics14232767

**Published:** 2024-12-09

**Authors:** Tianwei Liang, Yanhui Cui, Man Hu, Di Cao, Honggai Yan, Li Li

**Affiliations:** Beijing Children’s Hospital, Capital Medical University, Beijing 100070, China; liangtw_750913@vip.126.com (T.L.); cyhdoc@hotmail.com (Y.C.); human1125@163.com (M.H.); caoxiaodi_katherie@163.com (D.C.); yanhonggai@163.com (H.Y.)

**Keywords:** cat scratch disease, neuroretinitis, Bartonella henselae

## Abstract

Bartonella henselae is a Gram-negative bacillus, mainly parasitizing on cats. When a child is scratched by a cat, they may present with the disease symptoms including regional lymphadenopathy, malaise, fever, and splenomegaly, which is known as cat-scratch disease (CSD). Ocular manifestations occur in 5–10% of patients with CSD. Neuroretinitis is the most common, and, in addition, Parinaud oculoglandular syndrome, endophthalmitis, retinochoroiditis, vascular occlusions, multiple mass-like lesions resembling ocular metastases, serous macular detachments, and retinal vasoproliferative lesions may also occur. We report a case of unilateral exudative retinal detachment and uveitis with a large subretinal deposit on the macula in a 6-year-old female with CSD, along with lymphadenitis on her left thigh. To the best of our knowledge, this case of exudative retinal detachment and uveitis with a large subretinal deposit under the retina affecting the macular area above the optic disc has not been previously reported.

**Figure 1 diagnostics-14-02767-f001:**
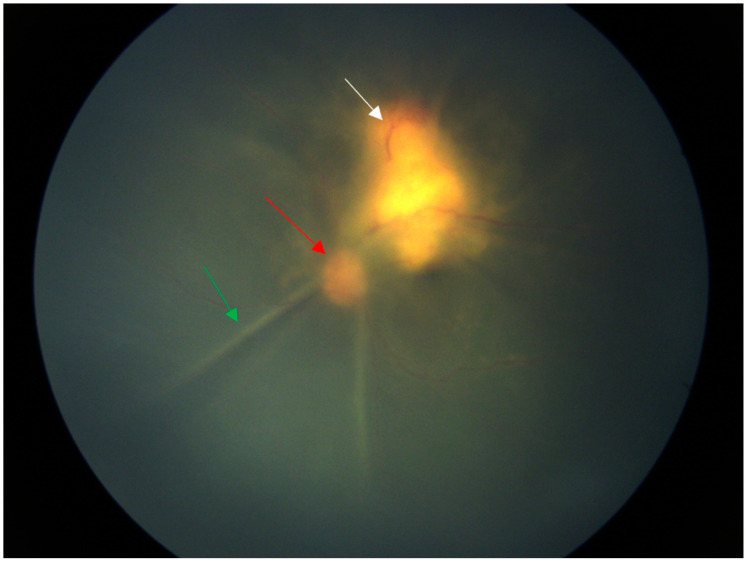
Fundus of a six-year-old Chinese girl with cat scratch disease. At the time of the patient’s visit for examination, the vision of the left eye was light perception, the vision of the right eye was 1.0, the intraocular pressure of the right eye was 17 mmHg, and the intraocular pressure of the left eye was 12 mmHg. There was corneal endothelial cell deposition on the left+, aqueous flare++, dilated pupil, clear lens, vitreous opacity++, total retinal shallow detachment. There was a huge yellow–white mass under the temporal superior retina adjacent to the optic disc. The mass involved the macular area. A month ago, a hard lump was found below the inguinal area on the inner side of her left thigh, accompanied by itching and tenderness. There was no rupture, swelling, fever or cough. Eighteen days ago, there was obvious conjunctival congestion in the left eye. The lump on the left thigh became enlarged and reddened. The skin temperature was high and there was tenderness but no rupture. The local hospital treated it as an infection (the medication was unclear). The parents unintentionally noticed a decrease in vision in the left eye, with only light perception. At the local hospital, ocular B-ultrasound showed an enhanced band-like echo in the dark area of the vitreous body of the left eye. B-ultrasound of the lymph nodes at the root of the left thigh showed enhanced local tissue echo. The white blood cell count was 17.78 × 10^9^/L. Considering infection, cephalosporin injection was given for 11 days without improvement. *We immediately took the vitreous humour for the metagenome next-generation sequencing (mNGS), and Bartonella henselae was detected as a result. Moreover, the pathological examination of the thigh nodule puncture showed the typical characteristics of Bartonella infection: the thigh nodule was soft tissue purulent inflammation, with special staining Warthin Starry (+) and Gram staining (-). Since the examination results were clear, no other tests were conducted.* The explanation of the arrows: The green arrow indicates the retinal folds caused by retinal detachment, the red arrow indicates the optic disc, and the white arrow indicates the huge yellow–white mass under the retina.

**Figure 2 diagnostics-14-02767-f002:**
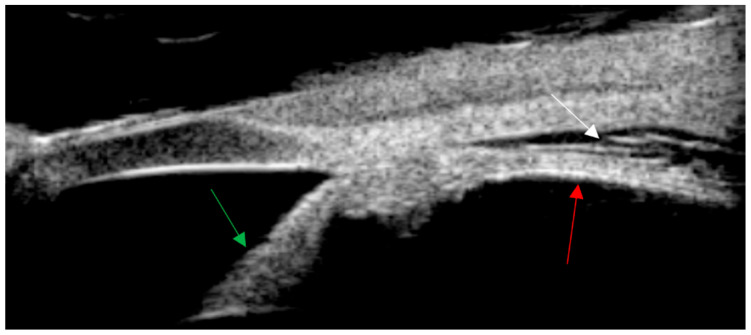
Ultrasound biomicroscopy (UBM) shows leakage in the supraciliary space of the left eye. The “explanation of the arrows: The red arrow indicates the continuation of the ciliary body flat part and the choroid. The green arrow indicates the iris. The white arrow indicates the supraciliary space.

**Figure 3 diagnostics-14-02767-f003:**
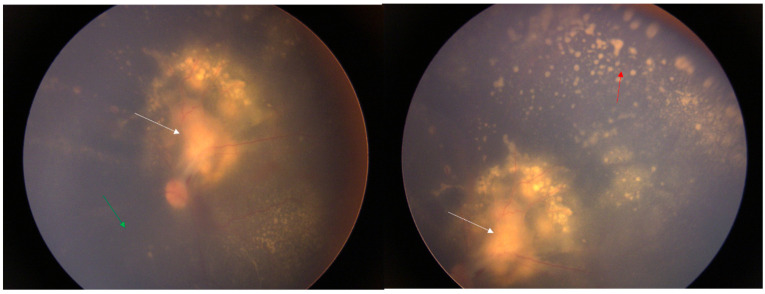
After 25 days of medication treatment, the boundary of the optic disc was clearer than before, and the retina was reattached. There was no significant change in the yellow–white exudate under the retina above the temporal side of the optic disc. Point-like exudates appeared under the peripheral retina, and the vitreous inflammation, was reduced. Reports from the literature indicate that rifampicin, ciprofloxacin, gentamicin, azithromycin, and compound sulfamethoxazole are all effective against cat scratch disease [1,2]. It is recommended that gentamicin be used preferentially for severe cases. In clinical treatment, antibiotics are usually used in combination. It is recommended to use drugs such as aminoglycosides, azithromycin, and quinolones. Clinical treatment generally requires more than 2 weeks. If necessary, lymph node puncture, pus aspiration, or resection in the affected area can be considered [1]. The effectiveness of steroids is uncertain. This patient was treated with rifampicin eye drops and erythromycin eye ointment topically in the eye, and rifampicin, azithromycin, and steroids systemically. The retinal detachment in the left eye was reattached, and the enlarged lymph nodes on the left thigh subsided. However, the huge exudate under the retina in the macular area could not be absorbed, resulting in a final visual acuity of only 0.03. The explanation of the arrows: The white arrow is the yellow–white mass under the retina, the green arrow is the flattened retina, and the red arrow is the yellow–white point-like exudate.

Cat scratch disease (CSD), also known as benign lymphohistiocytosis, is an infectious disease caused by cat or dog scratches or bites. The pathogen is Bartonella henselae. As the disease progresses, the initial lymphadenitis may develop into suppurative granulomatous lesions, with the formation of epithelioid cells, multinucleated giant cells and so on. It may also invade the optic nerve and the inside of the eye along with the blood, causing optic neuritis and retinitis [3]. Systemic symptoms precede ocular manifestations. Visual symptoms appear approximately one month after vaccination, and CSD (2/3) is the most common cause of optic neuritis, although only 1–2% of infected patients experience it [4].

Cat scratch disease presents with various manifestations in the eyes, including Parinaud’s oculoglandic syndrome, optic disc edema and macular stellate neuroretinitis (LISN), multifocal retinitis, focal and multifocal choroiditis, vascular occlusion and bacterial angiomatosis, uveitis, vitritis, exudative retinal detachment, etc. [5]. However, there are almost no reports of a large amount of subretinal exudate in the macular area, accompanied by retinal detachment and severe panuveitis, as described in this article [5]. Hao Hong et al. reported the first case of cat scratch disease with both uveitis and nodular lesions in the fundus, but without retinal detachment [1].

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
