# Peer review of "Unilateral Exudative Retinal Detachment and Uveitis Accompanied by a Large Subretinal Deposit on the Macula Secondary to Bartonella Henselae Infection"

_diagnostics, 2024, doi:10.3390/diagnostics14232767_

Round 1
Reviewer 1 Report
Comments and Suggestions for Authors
The abstract discusses a clinically significant topic—Cat-scratch disease (CSD) caused by Bartonella henselae and its ocular complications. It focuses on a single subject, which has not been previously reported, adding novelty and importance to the medical literature.
This case study can enhance the understanding of unusaual complications linked to CSD.
The abstract clearly outline the patient's symptom, diagnosis, and the unusual presentation (exudative retinal detachment and uveitis), providing a concise overview.
However, Since only one patient was involved in the study, broader clinical validation remains a challenge.
Please verify the spelling of medical terms.
Reviewer 2 Report
Comments and Suggestions for Authors
I appreciated the opportunity to review this paper that discusses a rare case of unilateral exudative retinal detachment and uveitis with a large subretinal deposit in the macular area, caused by Bartonella in a 6-year-old girl.
However, I do have some suggestions that I think could improve the paper:
-A brief explanation of the disease’s pathophysiology would enhance understanding for the reader
- Some sentences are overly complex, for example: "The mass involved the macular area, and one month after onset, the left eye showed congestion with decreased visual perception." This could be split into 2 sentences.
- It is unclear whether additional tests were performed to exclude other causes of uveitis and retinal detachment (serological tests for Toxoplasma ??? or PCR for common ocular pathogens???)
- Citations to official guidelines or systematic reviews on ocular manifestations of Bartonella should be included
